# CRISPR/Cas9-targeted mutagenesis of *OsERA1* confers enhanced responses to abscisic acid and drought stress and increased primary root growth under nonstressed conditions in rice

Takuya Ogata[1], Takuma Ishizaki[2], Miki Fujita[3], Yasunari Fujita[1,4]*

**1** Biological Resources and Post-harvest Division, Japan International Research Center for Agricultural Sciences (JIRCAS), Tsukuba, Ibaraki, Japan, **2** Tropical Agriculture Research Front (TARF), Japan International Research Center for Agricultural Sciences (JIRCAS), Ishigaki, Okinawa, Japan, **3** RIKEN Center for Sustainable Resource Science, Tsukuba, Ibaraki, Japan, **4** Graduate School of Life and Environmental Sciences, University of Tsukuba, Tsukuba, Ibaraki, Japan

* yasuf@affrc.go.jp

**Data Availability Statement:** All relevant data are within the manuscript and its Supporting Information files.

## Abstract

Abscisic acid (ABA) signaling components play an important role in the drought stress response in plants. *Arabidopsis thaliana* ENHANCED RESPONSE TO ABA1 (*ERA1*) encodes the β-subunit of farnesyltransferase and regulates ABA signaling and the dehydration response. Therefore, *ERA1* is an important candidate gene for enhancing drought tolerance in numerous crops. However, a rice (*Oryza sativa*) ERA1 homolog has not been characterized previously. Here, we show that rice *osera1* mutant lines, harboring CRISPR/Cas9-induced frameshift mutations, exhibit similar leaf growth as control plants but increased primary root growth. The *osera1* mutant lines also display increased sensitivity to ABA and an enhanced response to drought stress through stomatal regulation. These results illustrate that OsERA1 is a negative regulator of primary root growth under nonstressed conditions and also of responses to ABA and drought stress in rice. These findings improve our understanding of the role of ABA signaling in the drought stress response in rice and suggest a strategy to genetically improve rice.

## Introduction

The increased occurrence of extreme weather events due to climate change severely hinders crop production. Drought is a major abiotic stress limiting rice productivity in rainfed lowland rice agro-ecosystems [1, 2]. A number of physiological and molecular studies have revealed that phytohormone signaling pathways, such as those of abscisic acid (ABA), auxin, and brassinosteroids, play key roles in regulating the drought stress response in plants [3]. Of these, ABA signaling components function as central regulators in the drought stress response [4, 5]. ABA coordinates the plant's responses to decreased water availability. Cellular dehydration during

**Funding:** This work was supported in part by Japan Society for the Promotion of Science (JSPS) of Japan (Grants-in-Aid for Scientific Research (C) JP16K07412 to Y.F.) and the Ministry of Agriculture, Forestry and Fisheries (MAFF) of Japan. The funders had no role in study design, data collection and analysis, decision to publish, or preparation of the manuscript. URL of each funder website: JSPS: https://www.jsps.go.jp/english/e-grants/ MAFF: https://www.maff.go.jp/e/index.html

**Competing interests:** The authors have declared that no competing interests exist.

seed maturation and post-germination growth increases endogenous ABA levels, which triggers multiple developmental and physiological responses, including stomatal closure and the activation of dehydration-responsive gene expression [6]. Numerous genes involved in ABA signaling have been targeted in efforts to engineer improved drought tolerance [7].

The *Arabidopsis thaliana ENHANCED RESPONSE TO ABA1* (*ERA1*), which encodes the β-subunit of the protein farnesyltransferase, regulates ABA signaling and the dehydration response [8–10]. Protein farnesylation is a post-translational modification by which a farnesyl isoprenoid is attached to a C-terminal CaaX motif of the target protein, where "C" is the farnesylated cysteine, "a" is usually an aliphatic amino acid, and "X" is typically an alanine, cysteine, glutamine, methionine, or serine residue [11]. The farnesyl modification promotes protein association with membranes [11]. Among the approximately 700 *Arabidopsis thaliana* proteins identified as potential targets of ERA1-mediated farnesylation, CYP85A2, ASG2, and HSP40 proteins have been shown to function as negative regulators of ABA signaling [12–14]. CYP85A2 is a cytochrome P450 enzyme involved in brassinosteroid biosynthesis, whereas ASG2 is a WD40 protein implicated in seed germination and HSP40 functions as a molecular chaperone [12–14]. Another potential farnesylation target protein, AtNAP1;1, modulates cell proliferation and cell expansion during leaf development [15]. As such, ERA1 has been shown to have pleiotropic roles in different biological processes, such as defense against pathogens [16–18], heat-stress responses [14, 19, 20], hormonal responses [13, 21], and growth and development [22–25]. However, initially, the *era1* mutant was identified as a negative regulator of ABA signaling in seed germination [8]. In addition, *era1* mutant plants showed an enhanced response to ABA in stomatal closure [9, 10]. Therefore, *ERA1* is considered to be a candidate gene for increasing drought tolerance in a variety of crops [26–29].

The clustered regularly interspaced short palindromic repeats (CRISPR)/CRISPR-associated protein 9 (CRISPR/Cas9) system is a vital tool for editing the genomes of a wide range of organisms, including plants [30]. CRISPR/Cas9-mediated targeted mutagenesis is a crucial tool for functional analyses of plant genes and for crop improvement [31]. In this study, we used a CRISPR/Cas9-mediated genome editing approach to functionally characterize the role of ERA1 in plants. We showed that rice *osera1* mutant lines harboring CRISPR/Cas9-induced frameshift mutations in *OsERA1* display similar leaf growth as wild-type (WT) plants but enhanced primary root growth. The *osera1* mutant lines also exhibit an enhanced response to ABA and drought stress via stomatal regulation. These findings suggest that OsERA1 acts as a negative regulator of primary root growth under nonstressed conditions and as a negative regulator of responses to ABA and drought stress in rice.

## Materials and methods

### Plasmid construction and transformation

Target sites and single guide RNAs (gRNAs) for the first, third, and fifth exons of *OsERA1* (LOC_Os01g53600) adjacent to a protospacer-adjacent motif (PAM) were designed using the CRISPR-P design tool (http://crispr.hzau.edu.cn/CRISPR/) [32]. Each gRNA was designed to contain a restriction enzyme (RE) recognition site for the downstream PCR/RE-based selection steps [33]. To construct Cas9-gRNA plasmids, the primer pairs (S1 Table) were annealed and inserted into *Bsa*I sites of the *pCAMBIA1300*-based *pRGEB31* vector [34] (Addgene plasmid #51925). The resultant binary constructs, *pRGEB31-gRNA1*, *pRGEB31-gRNA2*, and *pRGEB31-gRNA3*, were introduced into *Agrobacterium tumefaciens* strain LBA4404, and used for transformation of immature rice (*Oryza sativa* L. ssp *japonica* cv. Nipponbare) embryos as described [35], except that a hygromycin concentration of 50 mg L$^{-1}$ was used. To confirm that the putative T$_0$ transformants harbored the transgene, genomic DNA was analyzed by

PCR using primers that amplify the *Cas9* fragment. PCR/RE assays were performed to detect mutations around the gRNA target sites as described [33] with minor modifications. Samples were PCR-amplified using Blend Taq-plus- (Toyobo, Osaka, Japan) and the products were digested with *Sac*I, *Taq*I, and *Hpy*166II for gRNA1, 2, and 3, respectively. The PCR products obtained from PCR/RE-positive $T_0$ events were subjected to direct sequencing. The genotypes of $T_0$ events were analyzed using DNA sequencing chromatograms. Then, *Cas9*-free $T_1$ plants were selected by PCR analysis using the *Cas9*-specific primer pair, and the PCR products amplified from the *Cas9*-free $T_1$ plants were used to determine the segregation of the targeted mutations by direct sequencing. All primers used in this study are listed in S1 Table.

## Plant growth conditions

$T_3$ and $T_4$ plants were subjected to physiological tests. After seven days of imbibition at 13˚C, rice seeds were germinated in plastic dishes with enough water for five days and then grown in a $CO_2$-regulated growth chamber (Biotron LH-410-S: Nippon Medical & Chemical Instruments, Osaka, Japan) to ensure that $CO_2$ concentrations did not fall below 400 ppm [36], under controlled conditions of 16 h light at 28˚C/8 h darkness at 25˚C and a light intensity of 150 µmol photons $m^{-2}$ $s^{-1}$. For ABA treatments, the seeds were incubated with 1 µM ABA one day after sowing for 5 days in 9 cm plastic petri dishes in the same temperature-controlled growth chamber.

## Mild drought stress treatment

Rice plants were grown in an isolation greenhouse under 28˚C day/24˚C night temperatures. The germinating seedlings were grown in soil-filled, open-bottomed small plastic tubes for 16 days. At 16 days after sowing, the plants were transferred to 1/5000a Wagner pots (17 cm diameter; 19 cm height) filled with 2.2 kg of mixed soil (1.56 kg of soil dry weight per pot) made by mixing Bonsol No.2 (Sumitomo Chemical, Osaka, Japan), which is an artificial granular cultivation soil, and red clay at a volume ratio of 1:1 with a soil water content of 29.5% (weight of moisture content/soil dry weight + weight of moisture content + plant weight). Twelve, ten, and twelve pots were prepared for the *osera1* mutant lines (M1G and M2T), and WT, respectively. The pots were covered with transparent polyethylene shower caps to prevent water loss. All pots were continually saturated with water for 10 days. At 27 days after sowing, the pots in the water-deficit stress (WS) treatment were drained of excess water overnight. The weight of individual pots was recorded every few days, starting on the day after draining. Experiments were designed based on the definition of total mass-based soil water content (%) that was calculated as [weight of moisture content/total soil weight (soil dry weight + weight of moisture content) + plant weight] × 100. In preliminary tests, we confirmed that plants were subjected to drought stress in WS pots with a soil water content of 40% (S1 Fig). As the water-holding capacity of the soil was 49%, the soil water content was adjusted to approximately 60% (water weight/total soil weight plus plant weight) for the well-watered (WW) treatment and 40% for the WS treatment to compensate for water loss due to transpiration at the time of weight measurement. A soil water content of 30%, 40%, and 49% was confirmed to be equivalent to a soil matric potential of –54 kPa, –11 kPa, 0 kPa, respectively, by measurements using a tensiometer (pressure gauge type DIK-8343, Daiki Rika Kogyo Co., Ltd., Saitama, Japan). Based on the results of our preliminary tests (S1 Fig) and a previous report [37], we determined that the soil water content was adjusted to 60% in WW pots and 40% in WS pots, respectively. The effect of the plant weight on soil water content could be disregarded during the experimental period, since the plant weight was much smaller than the total soil weight. Relative growth rates during the period from p to q days after sowing were calculated as [(HWSq–

HWSp)/(HWWq–HWWp)] × 100, where HWSp and HWSq are plant height in WS pots at p and q days after sowing, respectively, and HWWp and HWWq are plant height in WW pots at p and q days after sowing, respectively.

## Stomatal conductance measurement

Stomatal conductance (mol $H_2O$ $m^{-2}$ $s^{-1}$) was measured for the longest leaf of each plant with a LI-6400XT portable photosynthesis system (LI-COR Biosciences, Lincoln, NE, USA). As the ambient $CO_2$ concentration measured using the LI-6400XT was 480 ppm, the $CO_2$ concentration of the input flow, chamber block temperature, and intensity of the LED light source were set to 480 μmol $mol^{-1}$, 28.5°C, and 1,800 μmol $m^{-2}$ $s^{-1}$, respectively. Relative stomatal conductance rates during the period from p to q days after sowing were calculated as (CWSq/CWSp) × 100, where CWSp and CWSq are stomatal conductance in WS pots at p and q days after sowing, respectively.

## Statistical analysis

A one-way ANOVA with Dunnett's multiple-comparison test were performed using R software version 3.6.3 [38].

## Results

### CRISPR/Cas9-taregted mutagenesis of *OsERA1*

To understand the role of *OsERA1* in the plant's response to ABA and drought stress, we created rice *OsERA1* mutants using CRISPR/Cas9. *OsERA1* encodes two putative alternative transcripts, *OsERA1.1* and *OsERA1.2*, both of which consist of 14 exons (Fig 1A). To obtain *OsERA1* mutants, three gRNAs, gRNA1, gRNA2, and gRNA3, were designed to target the first, third, and fifth exons, respectively (Fig 1A).

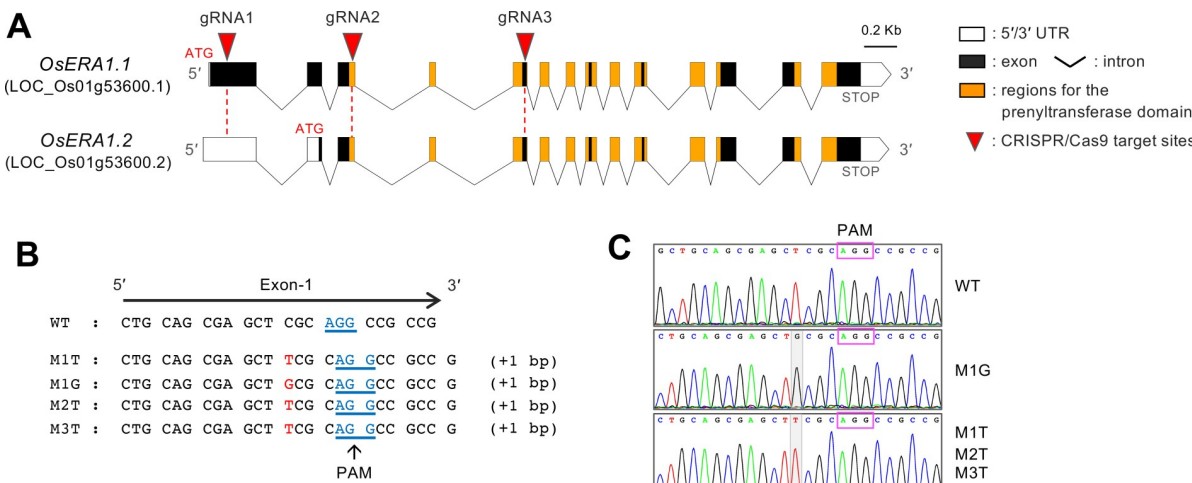

**Fig 1. CRISPR/Cas9-mediated mutagenesis of OsERA1 in rice.** (A) Schematic diagram of *OsERA1* and the target sites. *OsERA1* encodes two putative alternative transcripts, *OsERA1.1* and *OsERA1.2*. The guide RNAs (gRNAs) gRNA1, gRNA2, and gRNA3 were designed to target exon-1, -3, and -5, respectively, to induce mutations in the coding region of *OsERA1*. (B) The insertion mutations of alleles identified from sequence analysis of PCR amplicons from four *osera1* mutant lines, M1T, M1G, M2T, and M3T, with homozygous mutations at the gRNA1 target site. The inserted base (G/T) and protospacer-adjacent motif (PAM; boxed regions) are in red and blue font, respectively. (C) DNA sequence chromatogram of the homozygous T2 progenies of the four mutant lines containing a single nucleotide insertion (T or G) at the same position in the gRNA1 target site. Representative sequence data are shown. The inserted base (G/T) of the alleles are shaded in gray.

Three *OsERA1* CRISPR/Cas9 constructs containing each gRNA were used to transform the rice cultivar Nipponbare, which is widely used as a standard cultivar for studies of lowland rice. We generated 7, 201, and 65 $T_0$ transformants for the constructs containing gRNA1, gRNA2, and gRNA3, respectively, and identified multiple $T_0$ transformants harboring target mutations for all three constructs using PCR-based genotyping (S2 Table). However, we failed to obtain homozygous progeny plants with the desired mutations at the gRNA2 and gRNA3 target sites (S2 Table). For example, the growth of the homozygous progenies of the M4 and M5 mutant lines (S2 Table), M4-HM and M5-HM, with deletions at the gRNA2 target site was arrested at the plumule stage, whereas the heterozygous progenies of M4 and M5, M4-HT and M5-HT, grew normally (S2 Fig). Thus, mutations at the gRNA2 and gRNA3 target sites appear to be lethal. By contrast, four independent lines, M1T, M1G, M2T, and M3T, with homozygous mutations at the gRNA1 target site, were obtained (Fig 1B and S2 Table). Since the M1 line had a biallelic mutation in the $T_0$ generation, two independent alleles, M1T and M1G, segregated at subsequent generations. All four mutant alleles had a single nucleotide insertion (T or G) at the same position in the gRNA1 target site, inducing frameshift mutations that introduce premature stop codons in the *OsERA1.1* transcript (Fig 1B and 1C), whereas the mutant mRNAs may still have the ability to produce N-terminal truncated proteins with sequence similarity to OsERA1, except in the first 28 amino acids (S3 Table).

## The *osera1* mutants display increased primary root growth and sensitivity to ABA

Under nonstressed conditions, the seedlings of four *osera1* mutant lines, M1T, M1G, M2T, and M3T, showed similar leaf growth but significantly enhanced primary root growth compared with WT plants (Fig 2). $T_3$ and $T_4$ homozygous plants of the M1G and M2T mutant lines harboring a single nucleotide insertion (T or G) at the same position at the gRNA1 target site were selected for further study. The seedlings were treated with or without 1 µM ABA for five days to evaluate the effect of mutations at the gRNA1 target site on ABA sensitivity. The leaf and root growth of the M1G and M2T mutant lines were significantly more sensitive to ABA treatment than the WT (Fig 3), supporting the hypothesis that, like *Arabidopsis thaliana* ERA1 [8–10], OsERA1 is a negative regulator of ABA signaling. A recent study showed that the Arabidopsis *era1-2* mutant has a permeable cuticle [18]. In maize (*Zea mays* L.), cuticle-dependent leaf permeability is regulated by ABA signaling pathways [39]. We thus assessed the leaf cuticle permeability of rice *osera1* lines. No significant difference in cuticle permeability was observed between the *osera1* line and WT plants (S3 Fig).

## The *osera1* mutants show an enhanced response to mild drought stress

Since OsERA1 is involved in ABA signaling at the seedling stage, we analyzed the physiological responses of the M1G and M2T mutant lines to mild drought stress during later growth stages in a greenhouse under controlled temperature conditions (Fig 4A). Seedlings were subjected to water-deficit stress (WS) or control (well-watered, WW) treatment and their heights were compared at different time points. Although no substantial differences were observed in the relative growth rates of the WT and mutant seedlings at 27 to 35 days after sowing, the relative growth rates of the M1G and M2T plants at 35 to 44 days after sowing were significantly lower than those of WT plants (Fig 4B). These data suggest that M1G and M2T plants exhibit an enhanced response to drought stress in comparison with WT plants (Fig 4B).

Next, to examine the physiological response to drought stress in the *osera1* mutants, we measured the stomatal conductance of M1G and M2T mutant lines through the rate of gas exchange in the longest leaf of the individual plants. Soil water content was measured daily

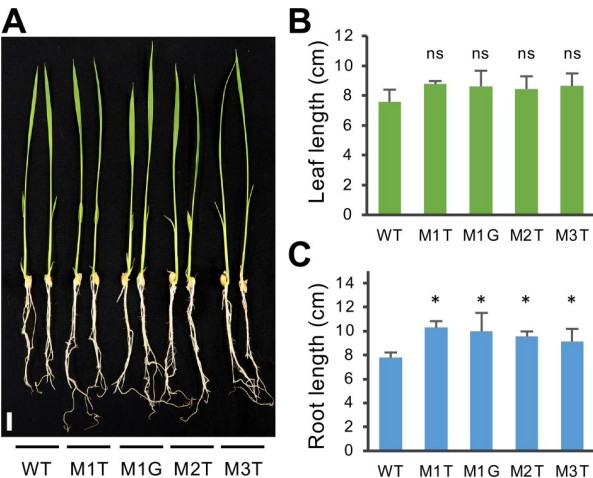

**Fig 2. The *osera1* homozygous mutant lines show similar leaf growth but significantly enhanced primary root growth.** (A) Growth phenotype of young seedlings of four *osera1* mutant lines, M1T, M1G, M2T, and M3T. Seeds were germinated and grown hydroponically for two weeks; representative plants are shown. Scale bar = 1 cm. Leaf length (B) and root length (C) of the seedlings were measured at nine days after germination. Values are presented as means ± SD ($n = 6$, $^*p < 0.05$ from WT by a one-way ANOVA with a Dunnett's multiple-comparison test and ns means no significance).

from 70 to 73 days after sowing (Fig 4C). At 70 and 73 days after sowing, soil water content was adjusted to 60% in WW pots and 40% in WS pots, respectively (Fig 4C). The relative stomatal conductance rates of M1G and M2T plants between 70 and 71 days after sowing were significantly reduced compared with those of the WT, whereas the stomatal conductance between 71 and 72 days after sowing did not differ between the mutant and WT lines (Fig 4D). Thus, the *osera1* mutant plants respond to drought stress more rapidly than WT plants through accelerated stomatal closure. These observations support the view that OsERA1 is a negative regulator of the plant's response to drought stress.

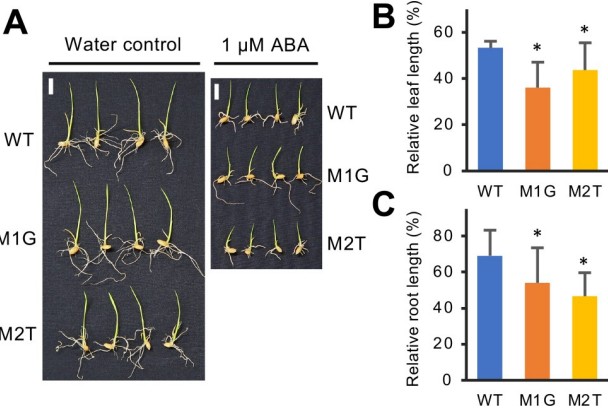

**Fig 3. The *osera1* mutant lines display enhanced sensitivity to ABA in leaf and root growth.** (A) Responses of M1G and M2T mutant lines to ABA solution. One-day-old germinated rice seeds were grown in 1 µM ABA solution in plastic dishes for five days. Photographs were taken at five days after the start of growth in 1 µM ABA solution; representative seedlings are shown. Scale bar = 1 cm. (B) Relative leaf length of the *osera1* mutant lines at five days after the start of growth in 1 µM ABA solution. (C) Relative root length of the *osera1* mutant lines at five days after the start of growth in 1 µM ABA solution. Values are presented as means ± SD ($^*p < 0.05$ from WT by a one-way ANOVA with a Dunnett's multiple-comparison test, $n = 10$).

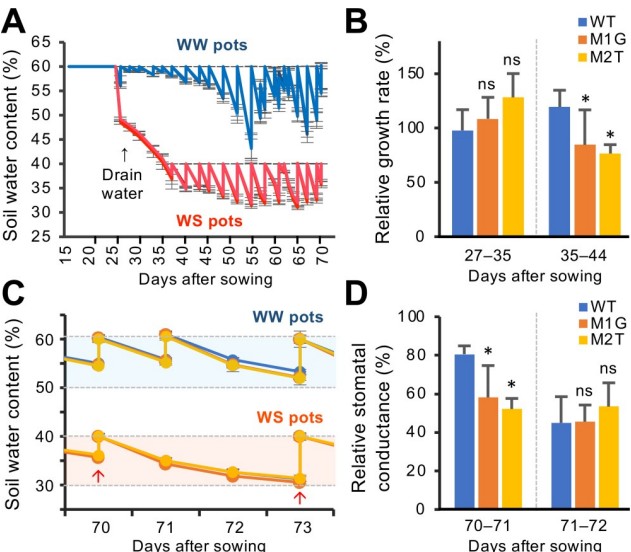

**Fig 4. The *osera1* mutant lines exhibit an enhanced response to drought stress.** (A) Soil water content (water weight/total soil weight). For water-deficit stress (WS) treatment, water was drained from the pots at 27 days after sowing and watering was withheld between 27 and 35 days after sowing. The soil water content was roughly maintained between 30% and 40% in WS pots from 35 days after sowing onwards, and between 50% and 60% in well-watered (WW) pots. (B) Relative growth rates of M1G and M2T mutant lines from 27 to 35 days and from 35 to 44 days after sowing. See Materials and Methods for details. (C) Variation of soil water content (% by weight). Soil water content was measured daily from 70 to 73 days after sowing. At 70 and 73 days after sowing, the soil water content was adjusted to 60% in WW pots and 40% in WS pots, respectively. Arrows indicate the time points at which water was supplied. Dashed lines indicate 30, 40, 50, and 60% of the soil water contents. Light blue and pale orange regions between dashed lines indicate the variation range of the soil water contents of WW and WS pots, respectively. (D) Relative stomatal conductance rates of M1G and M2T mutant lines from 70 to 71 days and from 71 to 72 days after sowing. In B and D, values are means ± SD ($n$ = 5 or 6, $^*p < 0.05$ from WT by a one-way ANOVA with a Dunnett's multiple-comparison test). See Materials and Methods for details.

## Discussion

Here, we demonstrated that the *osera1* mutant lines M1G and M2T display an enhanced response to ABA and drought stress via stomatal regulation (Figs 3 and 4), consistent with previous reports of ERA1 homologs in other plant species [26–29]. Thus, OsERA1 acts as a negative regulator of responses to ABA and drought stress in rice, suggesting that, like *Arabidopsis thaliana* ERA1 [8, 9], OsERA1 fulfills a key role in the drought stress response in rice. However, since the M1G and M2T *osera1* mutant lines have enhanced stomatal closure under drought stress (Fig 4B), these lines may have low yields and might thus not be suitable drought-tolerant lines for agricultural purposes. Considering that transgenic canola (*Brassica napus*) plants expressing *ERA1* antisense mRNAs driven by a drought-inducible promoter showed significantly improved drought tolerance under field conditions without negatively affecting yield [26], it may be worth exploring the effects of CRISPR/Cas9-targeted mutagenesis of the promoter region of *OsERA1*. Additionally, the M1G and M2T *osera1* mutations may be used to engineer improved drought tolerance in combination with other alleles that compensate for reduced yield [40].

The *osera1* mutant seedlings exhibited similar leaf growth but increased primary root growth under nonstressed conditions, implying that OsERA1 functions as a negative regulator of primary root growth under normal conditions. Although the relationship between *era1* mutation and lateral root growth have been reported in terms of ABA and auxin signaling via ABI3 [41, 42], the role of ERA1 in primary root growth was hitherto unknown. Low ABA concentrations promote primary root growth both by promoting the quiescence of the quiescent

center and suppressing stem cell differentiation in root meristems [43]. Furthermore, low ABA concentrations activate the HY5-ERF11 regulon, a two-tiered transcriptional cascade that represses the expression of genes involved in ethylene biosynthesis and thereby promotes primary root growth [44]. Thus, the increased primary root growth of the *osera1* mutant under nonstressed conditions may be due to an enhanced sensitivity to ABA. The basal role of ABA signaling under nonstressed conditions has been the focus of recent research [45, 46], and further research is required to clarify the roles of OsERA1 in this context.

This is the first report of CRISPR/Cas9-targeted mutagenesis being used to functionally characterize the role of ERA1 in plants. Although we designed three gRNAs for OsERA1 mutagenesis using the CRISPR-P design tool [32] to select appropriate target sites and avoid off-target effects, we were unable to obtain homozygous progeny plants with the desired mutations at two of the three gRNA target sites located in the first and fifth exons of the putative alternative transcripts, *OsERA1.1* and *OsERA1.2*, whose transcriptional start codons are in the first and second exons, respectively (Fig 1A and S2 Table). Considering that we generated four independent lines (M1T, M1G, M2T, and M3T) with the desired mutations at the gRNA1 target site located in the first exon of *OsERA1.1* and also in the 5′ UTR of *OsERA1.2* (Fig 1A and S2 Table), these data suggest that *OsERA1.2* may be essential for plant survival. This is consistent with previous reports showing that ERA1 has pleiotropic roles in different biological processes, including defense against pathogens [16–18], drought and heat-stress responses [8, 9, 14, 19, 20], hormonal responses [13, 21], and growth and development [22–25]. At the gRNA1 target site, we obtained only mutant lines with single insertion mutations (G or T) at the same nucleotide position (Fig 1A and S2 Table), causing frameshift mutations that create premature stop codons in the *OsERA1.1* transcript (Fig 1B and 1C). The mutant mRNAs may still retain the ability to produce N-terminal truncated proteins with sequence similarity to *OsERA1* except for the first 28 amino acids (S3 Table). Therefore, the lack of an N-terminal truncated region consisting of at least 28 amino acids may influence the mutant phenotype. Alternatively, the difference at the protein level might underlie the phenotype. However, since ERA1 has not yet been analyzed at the protein level, nor has an antibody been reported for this protein, more research is needed to determine why this mutation results in the observed phenotype. Thus, although ERA1 has been studied extensively using RNA interference, T-DNA insertion mutagenesis, and chemical mutagenesis [8–10, 26–29], our study using CRISPR/Cas9-targeted mutagenesis of *OsERA1* suggests the novel potential role of alternative transcripts and of a specific region of the gene in ERA1 function. Collectively, our findings improve our understanding of the role of ABA signaling in the drought stress response in rice and provide information that may be useful in efforts to genetically engineer drought tolerance in rice.

## Supporting information

**S1 Table. Primers used in this study.**
(XLSX)

**S2 Table. Genotypes of $T_0$ events with CRISPR/Cas9-induced mutations in *OsERA1*.**
(XLSX)

**S3 Table. Predicted OsERA1.1 amino acid sequences in *osera1* mutants.**
(XLSX)

**S1 Fig. Growth retardation and reppression of rice plants in the mild drought stress test.**
Representative photographs of WT plants exposed to mild drought stress for 58 days after sowing. Scale bars = 10 cm.
(PDF)

**S2 Fig. The growth of homozygous progenies of M4 and M5 mutant lines was arrested at the plumule stage.** (A) Deletions at the border of the third exon of alleles identified from a sequence analysis of PCR amplicons from the *osera1* mutant lines M4 and M5. The deleted bases of the alleles and the protospacer-adjacent motif (PAM) are in red and blue, respectively. (B) Growth phenotype of seedlings of the homozygous and heterozygous progenies of the M4 mutant lines, M4-HM and M4-HT, with deletions at the gRNA2 target site. Seeds were germinated and grown hydroponically for two weeks; representative plants are shown. Scale bar = 1 cm. (C) Growth phenotype of seedlings of the homozygous and heterozygous progenies of the M5 mutant lines, M5-HM and M5-HT, with deletions at the gRNA2 target site. Seeds were germinated and grown hydroponically for two weeks; representative plants are shown. Scale bar = 1 cm.
(PDF)

**S3 Fig. A cuticle permeability assay for the *osera1* lines.** Dye exclusion experiments were performed using leaves of 12-day-old seedlings as described by Cui *et al.* (2019). (A) Before the treatment. (B) Thirty minutes after immersion of the leaves in 0.05% (w/v) toluidine blue solution. No significant difference in cuticle permeability was observed between the leaves of the four *osera1* lines, M1T, M1G, M2T, and M3T, and the WT. Scale bars = 1 cm.
(PDF)

## Acknowledgments

We thank K. Shimizu, N. Hisatomi, I. Gejima, S. Shu, Y. Masamura, Y. Saito, and K. Ozawa for their excellent technical support, M. Toyoshima for skillful editorial assistance, and Y. Nagatoshi for critical reading of the manuscript.

## Author Contributions

**Conceptualization:** Takuya Ogata, Yasunari Fujita.

**Data curation:** Takuya Ogata, Takuma Ishizaki.

**Formal analysis:** Takuya Ogata.

**Funding acquisition:** Yasunari Fujita.

**Investigation:** Takuya Ogata, Takuma Ishizaki.

**Methodology:** Takuya Ogata, Miki Fujita.

**Project administration:** Yasunari Fujita.

**Resources:** Takuya Ogata, Takuma Ishizaki, Yasunari Fujita.

**Supervision:** Yasunari Fujita.

**Validation:** Takuya Ogata, Takuma Ishizaki, Yasunari Fujita.

**Visualization:** Takuya Ogata.

**Writing – original draft:** Takuya Ogata, Yasunari Fujita.

**Writing – review & editing:** Takuya Ogata, Yasunari Fujita.

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
