## [Decision Letter · Decision Letter 0]

25 Sep 2020

PONE-D-20-25974

CRISPR/Cas9-targeted mutagenesis of OsERA1 confers enhanced responses to abscisic acid and drought stress and increased primary root growth under nonstressed conditions in rice

PLOS ONE

Dear Dr. Fujita,

Thank you for submitting your manuscript to PLOS ONE. After careful consideration, we feel that it has merit but does not fully meet PLOS ONE’s publication criteria as it currently stands. Therefore, we invite you to submit a revised version of the manuscript that addresses the points raised during the review process.

We look forward to receiving your revised manuscript.

Kind regards,

Keqiang Wu, Ph.D

Academic Editor

PLOS ONE

Journal Requirements:

Reviewers' comments:

Reviewer's Responses to Questions

**Comments to the Author**

1. Is the manuscript technically sound, and do the data support the conclusions?

Reviewer #1: Yes

Reviewer #2: Partly

2. Has the statistical analysis been performed appropriately and rigorously? 

Reviewer #1: Yes

Reviewer #2: Yes

3. Have the authors made all data underlying the findings in their manuscript fully available?

Reviewer #1: Yes

Reviewer #2: Yes

4. Is the manuscript presented in an intelligible fashion and written in standard English?

Reviewer #1: Yes

Reviewer #2: No

5. Review Comments to the Author

Reviewer #1: The manuscript ” CRISPR/Cas9-targeted mutagenesis of OsERA1 confers enhanced responses to abscisic acid and drought stress and increased primary root growth under nonstressed conditions in rice” by Takuya Ogata et al, describe the making and characterization of rice era1 mutants.

I have a few major and minor comments:

Major comments:

1. The description of ERA1 protein function and era1 mutant phenotypes in Introduction is very short. The only role described role for ERA1 relates to ABA signaling. This is unfortunate as there is considerable information available for Arabidopsis era1, both mutant phenotypes and about protein function. For example, the Arabidopsis era1 mutant is defective in meristem and flower formation (http://www.plantcell.org/content/12/8/1267), pathogen responses (http://www.plantphysiol.org/content/148/1/348.short), heat stress (https://nph.onlinelibrary.wiley.com/doi/full/10.1111/nph.14212). The ERA1 protein has been placed to regulate farnesylation of an enzyme involved in brassinosteroid synthesis [https://www.nature.com/articles/nplants2016114]. Thus the results presented in the current manuscript that some of the obtained rice mutants are lethal (the gRNA2 and gRNA3 target sites) is not surprising as ERA1 regulates protein farnesylation which is involved in many different biological pathways. I suggest that the authors expand the Introduction to more broadly illustrate that ERA1 is doing much more than only regulate ABA signaling. This can also help to explain why some of the mutants were lethal.

2. The Arabidopsis era1 mutant has a permeable cuticle (https://academic.oup.com/jxb/article/70/20/5971/5536716). This would influence loss of water and drought responses. Have the authors considered testing if the rice era1 also has permeable cuticle? If rice era1 has permeable cuticle it might change the conclusions from some of the experiments. For example interpretation of stomatal conductance data might change if rice era1 mutants have permeable cuticles. The toluidine blue stain to test for cuticle permeability is a very easy experiment to do.

Minor comment:

3. The rice era1 mutant was sensitive to drought (Fig. 4). This is opposite to the Arabidopsis era1 phenotype (drought tolerant). This is not surprising as the Arabidopsis era1 mutant is highly pleiotropic and involved in many signaling pathways. However, similar to Introduction I am missing in the Discussion some text related to the many functions that ERA1 and protein farnesylation has in different plant signaling pathways. See also point 2 above – if rice era1 has permeable cuticles this could also be an explanation for why it is drought sensitive.

Reviewer #2: Major Concerns/Suggestions:

1) The assumption of single base insertion at gRNA1 site in “M1T, M1G, M2T, and M3T” led to premature termination of stop codon, and that is the cause of the phenotype appears to wrong due to the following reasons:

Singe base insertion at the site shown in “M1T, M1G, M2T, and M3T” lead to the premature stop codon after 45 amino acid. Whereas mutations in gRNA2 and gRNA3 will produce a longer proteins. The authors need to translate the mutant sequence and show the protein produced in each mutant. I have translated the “M1T, M1G, M2T, and M3T” mutant sequence and found that in reading frame 2, the mutant mRNA can produce a protein with complete homology to ERA1 except for the first 25 amino acid. Since mutants of gRNA2 and gRNA3, which produce longer protein than “M1T, M1G, M2T, and M3T” mutant, are unable to grow normally, the assumption that premature termination is cause of the phenotype in sRNA1 is wrong. A Western blotting may show what is the length of protein produced. I think that the protein produced from reading frame2 without the first 25 amino acid may be responsible for the phenotype. whether the the gain of function rather than the loss of function is responsible for the phenotype?

2) The single base insertion mutants are more sensitive to ABA (Fig 3) and drought stress (Figure 4). Earlier studies have shown that reduction in expression levels of ERA1 leads to enhanced drought tolerance (Plant Journal 43, 413–424; J Exp Bot. 2013 Mar; 64(5): 1381–1392). How this can be explained?

3) Line #98: “ ----- 550 ppm CO2, and a light intensity of 150” Currently ambient CO2 concentration if only 410 ppm? Why a higher CO2 was used?

4) Some problem with soil moisture content and stress imposition: Line #108-109: “red clay with a soil water content of 29.5% (water weight/total soil weight)” – Whether the “total soil weight” is total soil dry weight? Figure 4A. Please check the formula used for soil moisture content. The correct formula for soil moisture content % = (Soil water content/soil dry weight)*100

5) Line #113: It says “Water-holding capacity of the soil was 49%” This value at what soil matric potential? Why 60% for well watered (about 11% higher than the WHC) and Drought 40% (about 9% lower than the WHC) were selected? Whether 40% SMC was stress? What was the soil type and soil matric potential? Since for seed germination only 29.5% was used (Line #108-109), 40% may not be a stress.

6) Line #129-130: Why a CO2 concentration of 480 ppm was used? Currently ambient CO2 concentration if only 410 ppm?

Minor comments:

Line #113, 216, 237 and other places: Replace “water stress” with “water-deficit stress”

Line #131-134: “days after sowing were calculated as (CWSq/CWSp) × 100, where CWSp and CWSq are stomatal conductance in WS pots at p and q days after sowing, respectively, and CWWp and CWWq are stomatal conductance in WW pots at p and q days after sowing,” - in the formula CWWp and CWWq are not mentioned.

6. PLOS authors have the option to publish the peer review history of their article (what does this mean?). If published, this will include your full peer review and any attached files.

Reviewer #1: No

Reviewer #2: **Yes: **Viswanathan Chinnusamy

---

## [Author Response · Author response to Decision Letter 0]

5 Nov 2020

RESPONSE 1:

We thank the Editor and Reviewers for their valuable comments and suggestions to improve our manuscript. All the changes and additions made in response to the Editor’s and Reviewers’ comments are in red font in the revised manuscript.

The following minor errors have been changed: 

We replaced “enhance” with “improve” (Line 37); “Several” with “A number of” (Line 43); “resistance” with “tolerance” (Lines 52 and 303); and “farnesyltransferase” with “the protein farnesyltransferase” (Line 54), respectively.

COMMENT 2:

Reviewer #1: The manuscript ” CRISPR/Cas9-targeted mutagenesis of OsERA1 confers enhanced responses to abscisic acid and drought stress and increased primary root growth under nonstressed conditions in rice” by Takuya Ogata et al, describe the making and characterization of rice era1 mutants.

I have a few major and minor comments:

Major comments:

1. The description of ERA1 protein function and era1 mutant phenotypes in Introduction is very short. The only role described role for ERA1 relates to ABA signaling. This is unfortunate as there is considerable information available for Arabidopsis era1, both mutant phenotypes and about protein function. For example, the Arabidopsis era1 mutant is defective in meristem and flower formation (http://www.plantcell.org/content/12/8/1267), pathogen responses (http://www.plantphysiol.org/content/148/1/348.short), heat stress (https://nph.onlinelibrary.wiley.com/doi/full/10.1111/nph.14212). The ERA1 protein has been placed to regulate farnesylation of an enzyme involved in brassinosteroid synthesis [https://www.nature.com/articles/nplants2016114]. Thus the results presented in the current manuscript that some of the obtained rice mutants are lethal (the gRNA2 and gRNA3 target sites) is not surprising as ERA1 regulates protein farnesylation which is involved in many different biological pathways. I suggest that the authors expand the Introduction to more broadly illustrate that ERA1 is doing much more than only regulate ABA signaling. This can also help to explain why some of the mutants were lethal.

RESPONSE 2:

Thank you for your nice suggestion. Based on your suggestion, we have expanded the Introduction to include a description of ERA1 as a pleiotropic regulator (Lines 55–73, 78–79). 

COMMENT 3:

2. The Arabidopsis era1 mutant has a permeable cuticle (https://academic.oup.com/jxb/article/70/20/5971/5536716). This would influence loss of water and drought responses. Have the authors considered testing if the rice era1 also has permeable cuticle? If rice era1 has permeable cuticle it might change the conclusions from some of the experiments. For example interpretation of stomatal conductance data might change if rice era1 mutants have permeable cuticles. The toluidine blue stain to test for cuticle permeability is a very easy experiment to do.

RESPONSE 3:

Thank you for your interesting suggestion. Based on your comment, we have checked the cuticle permeability of osera1 lines using toluidine blue staining. There were no significant differences in cuticle permeability between the osera1 line and WT plants. We have added a new supplementary figure (S3 Fig) and corresponding text in the Results section (Lines 222–226). 

COMMENT 4:

Minor comment:

3. The rice era1 mutant was sensitive to drought (Fig. 4). This is opposite to the Arabidopsis era1 phenotype (drought tolerant). This is not surprising as the Arabidopsis era1 mutant is highly pleiotropic and involved in many signaling pathways. However, similar to Introduction I am missing in the Discussion some text related to the many functions that ERA1 and protein farnesylation has in different plant signaling pathways. See also point 2 above – if rice era1 has permeable cuticles this could also be an explanation for why it is drought sensitive.

RESPONSE 4:

Thank you for your helpful comment. As we describe in RESPONSE 6, our results show that the osera1 mutant plants respond to drought stress more rapidly than do WT plants through accelerated stomatal closure (Fig. 4), consistent with the hypersensitive ABA response seen in primary root length of osera1 mutant plants (Fig. 3). Therefore, our results are not inconsistent with those described in the previous papers mentioned by the reviewer. Considering your comment, it occurred to us that a more accurate description of our results would be “enhanced response to drought stress”, which we used in the title, rather than “sensitive to drought stress”. We have thus altered the descriptions of our results throughout our manuscript (Lines 34, 82, 246, 256–257, 271, 290–291, and 296). In addition, we have added a sentence discussing our results (Lines 328–332). 

COMMENT 5:

Reviewer #2: Major Concerns/Suggestions:

1) The assumption of single base insertion at gRNA1 site in “M1T, M1G, M2T, and M3T” led to premature termination of stop codon, and that is the cause of the phenotype appears to wrong due to the following reasons:

Singe base insertion at the site shown in “M1T, M1G, M2T, and M3T” lead to the premature stop codon after 45 amino acid. Whereas mutations in gRNA2 and gRNA3 will produce a longer proteins. The authors need to translate the mutant sequence and show the protein produced in each mutant. I have translated the “M1T, M1G, M2T, and M3T” mutant sequence and found that in reading frame 2, the mutant mRNA can produce a protein with complete homology to ERA1 except for the first 25 amino acid. Since mutants of gRNA2 and gRNA3, which produce longer protein than “M1T, M1G, M2T, and M3T” mutant, are unable to grow normally, the assumption that premature termination is cause of the phenotype in sRNA1 is wrong. A Western blotting may show what is the length of protein produced. I think that the protein produced from reading frame2 without the first 25 amino acid may be responsible for the phenotype. whether the the gain of function rather than the loss of function is responsible for the phenotype?

RESPONSE 5:

Thank you for pointing this out. Based on your suggestion, we have translated the “M1T, M1G, M2T, and M3T” mutant sequence in silico and confirmed that the mutant mRNA can indeed produce a protein with homology to ERA1 except for the first 28 amino acids (S3 Table). This suggests that the N-terminal truncated region may influence the phenotype. In addition, as pointed out, the difference at the protein level may also affect the phenotype. However, ERA1 has not been analyzed at the protein level, nor have its antibodies been reported. Therefore, more research is needed to determine why this mutation results in the observed phenotype. In the revised version of our manuscript, we have thus added a new supplementary table (S3 Table), phrases (Lines 194–196), and sentences (Lines 332–342) to explain the current situation and have altered a sentence for clarity (Lines 328–329, 342–346). In addition, we have removed “that cause premature stop codons” in the Introduction (Line 80). 

COMMENT 6:

2) The single base insertion mutants are more sensitive to ABA (Fig 3) and drought stress (Figure 4). Earlier studies have shown that reduction in expression levels of ERA1 leads to enhanced drought tolerance (Plant Journal 43, 413–424; J Exp Bot. 2013 Mar; 64(5): 1381–1392). How this can be explained? 

RESPONSE 6:

Thank you for raising this good point. As we described in RESPONSE 4, our results show that the osera1 mutant plants respond to drought stress more rapidly than do WT plants through accelerated stomatal closure (Fig. 4), consistent with the hypersensitive ABA response seen in primary root length of osera1 mutant plants (Fig. 3). Therefore, our results are not inconsistent with those described in the previous papers mentioned by the reviewer. Upon further consideration, it occurred to us that a more accurate description of our results would be “enhanced response to drought stress”, which we used in the title, rather than “sensitive to drought stress”. Accordingly, we have altered the descriptions of our results throughout our manuscript (Lines 34, 82, 246, 256–257, 271, 290–291, and 296). 

COMMENT 7:

3) Line #98: “ ----- 550 ppm CO2, and a light intensity of 150” Currently ambient CO2 concentration if only 410 ppm? Why a higher CO2 was used?

RESPONSE 7:

Thank you for pointing this out. Our group (Nagatoshi et al., 2019) and other independent groups (Ohnishi et al. 2011 PCP 52:1249–1257; Tanaka et al. 2016 Breed Sci. 66:542–551) have shown that CO2 concentrations inside growth chambers that comprise a closed and limited space are much lower than those of the ambient atmosphere during the growth phase of rice and soybean, and that CO2 supplementation enhances rice and soybean growth in plant growth chambers. In our case, by setting the CO2 concentration to 550 ppm, the actual CO2 concentration inside the chamber used to grow rice and soybean plants was maintained above 400 ppm. We have altered this sentence to convey this information and have added a reference in the Materials and Methods section (Lines 115–118). 

COMMENT 8:

4) Some problem with soil moisture content and stress imposition: Line #108-109: “red clay with a soil water content of 29.5% (water weight/total soil weight)” – Whether the “total soil weight” is total soil dry weight? Figure 4A. Please check the formula used for soil moisture content. The correct formula for soil moisture content % = (Soil water content/soil dry weight)*100

RESPONSE 8:

Thank you for your comment. In this paper, “total soil weight” does not mean total soil dry weight but soil dry weight plus the weight of the moisture content. Therefore, the formula for soil water content (%) was calculated as [weight of moisture content/total soil weight (soil dry weight plus weight of moisture content) + plant weight] × 100. To use round numbers in this study, the values calculated on a total mass basis are presented as soil water contents; however, we have also added a dry weight value that readers can use to calculate the dry weight-based soil water content. We have provided clear definitions of these terms to avoid confusion (Lines 125–130; 136–140). 

COMMENT 9:

5) Line #113: It says “Water-holding capacity of the soil was 49%” This value at what soil matric potential? Why 60% for well watered (about 11% higher than the WHC) and Drought 40% (about 9% lower than the WHC) were selected? Whether 40% SMC was stress? What was the soil type and soil matric potential? Since for seed germination only 29.5% was used (Line #108-109), 40% may not be a stress.

RESPONSE 9:

Thank you for raising this question. We measured the soil matric potential at the water-holding capacity of the soil using a tensiometer. Our measurements confirmed that a soil water content of 30%, 40%, and 49% was equivalent to a soil matric potential of –54 kPa, –11 kPa, and 0 kPa, respectively. We used mixed soil, made by combining Bonsol No.2 (Sumitomo Chemical, Osaka, Japan), which is an artificial granular cultivation soil, and red clay at a volume ratio of 1:1. We have added sentences to provide this new information (Lines 125–130; 143–148). 

In preliminary tests, we confirmed that plants were subjected to drought stress in WS pots with a soil water content of 40% (S1 Fig). Based on the results of our preliminary tests (S1 Fig) and a previous report (Singh et al., 2007), we determined that the soil water content was adjusted to 60% in WW pots and 40% in WS pots, respectively. In pots with a soil water content of 60%, about 1–2 cm of water accumulates on the soil surface. The weight of individual pots was recorded every few days, and the soil water content was adjusted to approximately 60% for the WW treatment and 40% for the WS treatment to compensate for water loss due to transpiration at the time of weight measurement. The soil water content was roughly maintained between 30% and 40% in WS pots from 35 days after sowing onwards, and between 50% and 60% in WW pots. Therefore, the water content in WW pots was usually maintained above the water-holding capacity during the test. As shown in Fig. 4A, since the plants are grown in sunlight, the temperature rose depending on the weather and, consequently, the soil water content occasionally dipped below the water-holding capacity. We have added these data (S1 Fig) and the corresponding descriptions (Lines 146–148). 

Seeds were germinated in plastic dishes with enough water for 5 days and then the germinated seedlings were transferred to soil-filled plastic tubes as described in Lines 125–126. For clarity, we have added descriptions of this procedure in the Materials and Methods section (Line 114–115). 

COMMENT10:

6) Line #129-130: Why a CO2 concentration of 480 ppm was used? Currently ambient CO2 concentration if only 410 ppm?

RESPONSE 10:

Thank you for raising these questions. For the experiments, we set a CO2 concentration of LI-6400XT at 480 ppm, since 480 ppm was the ambient CO2 concentration monitored by LI-6400XT in the greenhouse where we grew our plants. We have added text to explain this point (Lines 159–160). 

COMMENT 11:

Minor comments:

Line #113, 216, 237 and other places: Replace “water stress” with “water-deficit stress”

Line #131-134: “days after sowing were calculated as (CWSq/CWSp) × 100, where CWSp and CWSq are stomatal conductance in WS pots at p and q days after sowing, respectively, and CWWp and CWWq are stomatal conductance in WW pots at p and q days after sowing,” - in the formula CWWp and CWWq are not mentioned.

RESPONSE 11:

Thank you for your suggestions. In response to your comment, we have replaced “water stress” with “water-deficit stress” (Lines 134, 251, and 272). In addition, we have removed the unneeded phrase (Lines 162–164).

---

## [Decision Letter · Decision Letter 1]

20 Nov 2020

CRISPR/Cas9-targeted mutagenesis of OsERA1 confers enhanced responses to abscisic acid and drought stress and increased primary root growth under nonstressed conditions in rice

PONE-D-20-25974R1

Dear Dr. Fujita,

We’re pleased to inform you that your manuscript has been judged scientifically suitable for publication and will be formally accepted for publication once it meets all outstanding technical requirements.

Kind regards,

Keqiang Wu, Ph.D

Academic Editor

PLOS ONE

Additional Editor Comments (optional):

Reviewers' comments:

Reviewer's Responses to Questions

**Comments to the Author**

1. If the authors have adequately addressed your comments raised in a previous round of review and you feel that this manuscript is now acceptable for publication, you may indicate that here to bypass the “Comments to the Author” section, enter your conflict of interest statement in the “Confidential to Editor” section, and submit your "Accept" recommendation.

Reviewer #1: All comments have been addressed

Reviewer #2: All comments have been addressed

2. Is the manuscript technically sound, and do the data support the conclusions?

Reviewer #1: Yes

Reviewer #2: Partly

3. Has the statistical analysis been performed appropriately and rigorously? 

Reviewer #1: Yes

Reviewer #2: Yes

4. Have the authors made all data underlying the findings in their manuscript fully available?

Reviewer #1: Yes

Reviewer #2: Yes

5. Is the manuscript presented in an intelligible fashion and written in standard English?

Reviewer #1: Yes

Reviewer #2: Yes

6. Review Comments to the Author

Reviewer #1: In the revised manuscript, the authors have answered all of my previous comment. I have only one additional comment:

1. In the revised version, lines 192 - 196 , there is a description of the effect of the Crispr mutations on the OsERA1.1 transcript and corresponding protein. This was very difficult to understand, and would benefit from careful editing. As far as I can tell from S3 Table, the crispr mutations leads to a truncated protein - if translation is initiated from the first ATG. However, if translation is started at the second ATG, the only effect is a somewhat shorter protein. If this is a correct interpretation, then I suggest to edit the text on lines 192-196 to make this easier to understand. For example, instead of trying to have all this information in one long sentence, it could be edited to several shorter sentences.

Reviewer #2: Most concerns were addressed.

1) The statistical significant in Fig 3 and and in few bars of Figure 4 may be checked as the error bars indicate the different may not be significant in few cases.

2) Relative stomatal conductance is also not a good measure of drought stress response as even in the well watered plants, stomatal conductance will change as the VPD changes.

7. PLOS authors have the option to publish the peer review history of their article (what does this mean?). If published, this will include your full peer review and any attached files.

Reviewer #1: No

Reviewer #2: No

---

## [Editor Report · Acceptance letter]

25 Nov 2020

PONE-D-20-25974R1 

CRISPR/Cas9-targeted mutagenesis of *OsERA1* confers enhanced responses to abscisic acid and drought stress and increased primary root growth under nonstressed conditions in rice 

Dear Dr. Fujita:

I'm pleased to inform you that your manuscript has been deemed suitable for publication in PLOS ONE. Congratulations! Your manuscript is now with our production department. 

Kind regards, 

on behalf of

Professor Keqiang Wu 

Academic Editor

PLOS ONE